# Submerged and Solid-State Fermentation of Spirulina with Lactic Acid Bacteria Strains: Antimicrobial Properties and the Formation of Bioactive Compounds of Protein Origin

**DOI:** 10.3390/biology12020248

**Published:** 2023-02-03

**Authors:** Ernesta Tolpeznikaite, Vadims Bartkevics, Anna Skrastina, Romans Pavlenko, Modestas Ruzauskas, Vytaute Starkute, Egle Zokaityte, Dovile Klupsaite, Romas Ruibys, João Miguel Rocha, Elena Bartkiene

**Affiliations:** 1Institute of Animal Rearing Technologies, Faculty of Animal Sciences, Lithuanian University of Health Sciences, Mickeviciaus Str. 9, LT-44307 Kaunas, Lithuania; 2Institute of Food Safety, Animal Health and Environment “BIOR”, Lejupes iela 3, Zemgales priekšpilsēta, Riga LV-1076, Latvia; 3Department of Anatomy and Physiology, Faculty of Veterinary, Lithuanian University of Health Sciences, Mickeviciaus Str. 9, LT-44307 Kaunas, Lithuania; 4Institute of Microbiology and Virology, Faculty of Veterinary, Lithuanian University of Health Sciences, Mickeviciaus Str. 9, LT-44307 Kaunas, Lithuania; 5Department of Food Safety and Quality, Faculty of Veterinary, Lithuanian University of Health Sciences, Mickeviciaus Str. 9, LT-44307 Kaunas, Lithuania; 6Institute of Agricultural and Food Sciences, Agriculture Academy, Vytautas Magnus University, K. Donelaicio Str. 58, LT-44244 Kaunas, Lithuania; 7Laboratory for Process Engineering, Environment, Biotechnology and Energy, Faculty of Engineering, University of Porto, 4200-465 Porto, Portugal; 8Associate Laboratory in Chemical Engineering, Faculty of Engineering, University of Porto, 4200-465 Porto, Portugal

**Keywords:** spirulina, L-glutamic acid, gamma-aminobutyric acid, biogenic amines, fermentation, lactic acid bacteria (LAB)

## Abstract

**Simple Summary:**

Spirulina (Arthrospira platensis) is an edible blue-green alga that shows many desirable physiological activities in humans and animals. In this study, we hypothesized that the Spirulina composition can be improved (by increasing the gamma-aminobutyric acid concentration) during biotreatment with selected lactic acid bacteria (LAB) strains. Fermentation is the most popular and typically economically effective solution in the food and feed industry and used as biotechnology for the bioconversion of materials to higher-added-value products. However, in addition to desirable compounds, LAB are involved in the processes of biogenic amine formation. This study showed that most of the fermented Spirulina samples possess exceptional antimicrobial activity against Staphylococcus. However, the ratios of biogenic amine/gamma-aminobutyric acid and biogenic amine/L-glutamic acid ranged from 0.5 to 62 and from 0.31 to 10.7, respectively. It was concluded that the formation of non-desirable compounds (biogenic amines) must also be considered due to the similar mechanism of their synthesis as well as the possibility of obtaining high concentrations in the end products.

**Abstract:**

The aim of this study was to investigate the changes in bioactive compounds (L-glutamic acid (L-Glu), gamma-aminobutyric acid (GABA) and biogenic amines (BAs)) during the submerged (SMF) and solid-state (SSF) fermentation of Spirulina with lactobacilli strains (*Lacticaseibacillus paracasei* No. 244; *Levilactobacillus* brevis No. 173; *Leuconostoc mesenteroides* No. 225; *Liquorilactobacillus uvarum* No. 245). The antimicrobial properties of the untreated and fermented Spirulina against a variety of pathogenic and opportunistic strains were tested. The highest concentrations of L-Glu (3841 mg/kg) and GABA (2396 mg/kg) were found after 48 h of SSF with No. 173 and No. 244 strains, respectively. The LAB strain used for biotreatment and the process conditions, as well as the interaction of these factors, had statistically significant effects on the GABA concentration in Spirulina (*p* ≤ 0.001, *p* = 0.019 and *p* = 0.011, respectively). In all cases, the SSF of Spirulina had a higher total BA content than SMF. Most of the fermented Spirulina showed exceptional antimicrobial activity against *Staphylococcus aureus* but not against the other pathogenic bacteria. The ratios of BA/GABA and BA/L-Glu ranged from 0.5 to 62 and from 0.31 to 10.7, respectively. The GABA content was correlated with putrescine, cadaverine, histamine, tyramine, spermidine and spermine contents. The L-glutamic acid concentration showed positive moderate correlations with tryptamine, putrescine, spermidine and spermine. To summarize, while high concentrations of desirable compounds are formed during fermentation, the formation of non-desirable compounds (BAs) must also be considered due to the similar mechanism of their synthesis as well as the possibility of obtaining high concentrations in the end products.

## 1. Introduction

*Arthrospira platensis* is an edible blue-green alga that shows beneficial activities in humans and animals [1]. Spirulina is cultivated worldwide as a fundamental ingredient in many nutraceutical formulations [2]. This alga contains high protein content, which includes all essential amino acids. Additionally, it contains valuable essential fatty acids, minerals, pigments, carotenoids and vitamins [3]. The probiotic and antioxidant properties of Spirulina have been widely reported [3,4,5]; therefore, Spirulina is used as a nutritional supplement in the human diet, as well as for animal nutrition, just to prevent gut dysbiosis and pathogen colonization [3]. The United States Food and Drug Administration (FDA) granted Spirulina the “Generally Recognized as Safe (GRAS)” status [4]. Moreover, Spirulina is a safe ingredient when grown under controlled conditions [4,6,7,8,9,10]. There is scientific evidence attesting to Spirulina’s hypolipemic, antihypertensive, antidiabetic, neuroprotective, antianemic, anticarcinogenic, hepatoprotective, antibacterial, antiviral and immunomodulatory properties [7,9,10,11,12]. In this study, we hypothesized that the Spirulina composition can be improved during biotreatment with selected lactic acid bacteria (LAB) strains. Fermentation is the most popular and typically economically effective solution in the food and feed industry and is used as biotechnology for the bioconversion of materials to higher-added-value products. Solid-state fermentation (SSF) consists of microbial growth and product formation on solid particles in the absence of water. This technology is more economical compared with the traditional method of biomass cultivation in a liquid medium containing nutrients.

Biotreatment/biotransformation with LAB is a popular solution to degrade plant and cyanobacterial cell walls and to produce smaller molecules with enhanced (immunomodulatory, antioxidant, antimicrobial, etc.) properties [13,14,15]. Additionally, via peptide bond hydrolysis, LAB proteases yield bioactive peptides with multiple health benefits [16].

Recently, the production of amino acids via a sustainable microbial approach (fermentation or enzymatic treatment) has gained interest [17,18]. However, the use of genetically modified microorganisms has been a major concern in the food and feed sectors [19]. This has led to the search for new (bio)technological starters. It was reported that wild-type LAB have the potential for the synthesis of various amino acids [20,21]. Lactic acid bacteria show economic advantages at the industrial scale and are generally recognized as safe microorganisms [17]. However, LAB multiplication in an environment that contains inorganic nitrogen is poor. Additionally, they often require an exogenous supply of nutrients (peptides and amino acids) to ensure their viability [22]. Many studies have concluded that the proteolytic system of LAB is important in the utilization of both proteins and peptides, and this enzymatic system activity can be designed by modeling the environmental and growth conditions [20,21,23,24].

Another compound that can be formed during protein metabolism is gamma-aminobutyric acid (GABA). Usually, GABA is enzymatically produced from L-glutamic acid (L-Glu) by glutamate decarboxylase [20]. This compound (GABA) has multiple physiological functions [20,24,25,26]. It was reported that many types of microorganisms can synthesize GABA [27,28], and LAB are very good candidates for GABA production. Additionally, LAB can excrete various antimicrobial compounds to the fermentable substrate/medium [29], simply by improving the multifunctional properties of the fermentable substrate. Although many LAB strains have been identified as good GABA producers, this process is strain-specific. Additionally, the specific processing conditions are important in this synthesis. Therefore, optimizing the technological conditions has become a very important approach for effective GABA synthesis.

In addition to desirable compounds, LAB are involved in the processes of biogenic amine (BA) formation. Biogenic amines are involved in several pathogenic syndromes [30]. However, their toxicity is related to the type of BA and the individual sensitivity of the person [31]. The most toxic BAs are tyramine (TYR) and histamine (HIS) [32,33]. However, the presence of 2-phenylethylamine (PHE), putrescine (PUT), cadaverine (CAD), agmatine (AGM), spermine (SPRM) and spermidine (SPRD) can lead to toxicity, because these BAs can potentiate the effects of histamine and tyramine toxicity [34]. Finally, high concentrations of BAs can have toxicological consequences for both humans and animals.

The aim of this study was to investigate the changes in bioactive compounds of proteinaceous origin (L-glutamic acid, GABA and BAs) in the submerged (SMF) and solid-state (SSF) fermentation of Spirulina with lactobacilli strains (*Lacticaseibacillus paracasei* No. 244; *Levilactobacillus brevis* No. 173; *Liquorilactobacillus uvarum* No. 245) and *Leuconostoc mesenteroides* No. 225. Taking into consideration that these strains previously showed a broad spectrum of antimicrobial activities, the antimicrobial properties of untreated (non-fermented) and fermented Spirulina against a variety of pathogenic and opportunistic strains (*Staphylococcus aureus, Escherichia coli, Acinetobacter baumannii, Staphylococcus haemolyticus, Salmonella enterica, Bacillus cereus*, *Proteus mirabilis, Klebsiella pneumoniae, Enterococcus faecium* and *Pseudomonas aeruginosa*) were tested.

## 2. Materials and Methods

### 2.1. Spirulina, Microorganisms and Algae Fermentation Conditions

Lyophilized Spirulina (*Arthrospira platensis*) powder (in 100 g: total carbohydrates 30.3 g, proteins 60.6 g, Na 1.1 g, Ca 151.5 mg, K 1.7 mg, Fe 48.5 mg) was purchased from Now Foods Company (Bloomingdale, IL, USA).

Characteristics of the used LAB strains (*Lacticaseibacillus paracasei* No. 244; *Levilactobacillus brevis* No. 173; *Leuconostoc mesenteroides* No. 225; *Liquorilactobacillus uvarum* No. 245) are reported by Bartkiene et al. [29].

The experimental design used in the current study is schematized in Figure 1.

For SMF, Spirulina powder was mixed with sterilized water in a ratio of 1:20 *w*/*w*, and for SSF, the Spirulina/water ratio was 1:2 *w*/*w*. The LAB strains were multiplied in MRS (De Man, Rogosa, and Sharpe) broth culture medium (Biolife, Milano, Italy) at 30 °C under anaerobic conditions for 24 h. A total of 3 mL of multiplied LAB [9.0 log_10_ CFU/mL] was inoculated in 100 mL of Spirulina. Afterward, the Spirulina samples were fermented under anaerobic conditions in a chamber incubator (Memmert GmbH Co. KG, Schwabach, Germany) for 24 and 48 h at 30 °C. Non-fermented samples were analyzed as controls. Before the analysis, non-fermented Spirulina was mixed with sterilized water in appropriate proportions for SMF and SSF conditions.

### 2.2. Evaluation of pH and Lactic acid Bacteria (LAB) Counts in the Spirulina Samples

The pH of Spirulina samples was evaluated with a pH meter (Inolab 3, Hanna Instruments, Villafranca Padovana, Italy) by inserting the pH electrode into the algal samples.

For the evaluation of LAB counts (log_10_ CFU/mL) in Spirulina samples, MRS agar (CM0361, Oxoid, Basingstoke, UK) was used.

### 2.3. Evaluation of the Concentration of L-Glutamic Acid (L-Glu) and Gamma-Aminobutyric (GABA) Acid in Spirulina Samples

The evaluation of the concentrations of L-Glu and GABA acid in Spirulina samples was performed on a TSQ Quantiva MS/MS coupled to a Thermo Scientific Ultimate 3000 HPLC instrument (Thermo Scientific, Waltham, MA, USA). Analysis is given in detail in Appendix A.

### 2.4. Evaluation of the Concentration of Biogenic Amines (BAs) in Spirulina Samples

The determination of the BAs in Spirulina was conducted using the method of Ben-Gigirey et al. (1998) [35], with some modifications (described in Appendix A).

### 2.5. Evaluation of the Antimicrobial Activity of Spirulina Samples

All algal samples were assessed for their antimicrobial activities against a variety of pathogenic and opportunistic wild bacterial strains previously isolated from humans and animals in the Lithuanian University of Health Sciences (*Staphylococcus aureus, Escherichia coli, Acinetobacter baumannii, Staphylococcus haemolyticus, Salmonella enterica, Bacillus cereus*, *Proteus mirabilis, Klebsiella pneumoniae, Enterococcus faecium* and *Pseudomonas aeruginosa*) by using the agar well-diffusion method.

For the agar well-diffusion assay, suspensions of 0.5 McFarland standard of each pathogenic bacterial strain were inoculated onto the surface of cooled Mueller–Hinton agar (Oxoid, Basingstoke, UK) using sterile cotton swabs. Wells with 6 mm diameters were punched in the agar and filled with 50 µL of the Spirulina samples (mixture of Spirulina powder and sterilized water). The antimicrobial activities against the tested bacteria were established by measuring the inhibition zone diameters (mm). The experiments were repeated three times, and the average diameter of the inhibition zones was calculated.

### 2.6. Statistical Analysis

The biotreatment of Spirulina was performed in duplicate, and all analytical experiments were carried out in triplicate. To evaluate the potential influences of different factors (SMF or SMF conditions, duration of fermentation, type of LAB strain used for fermentation) and their interactions on sample characteristics, data were compared using Duncan’s multiple range test with significance defined at *p* ≤ 0.05 using the IBM SPSS Statistics for Windows, v28.0.1.0 (142) (SPSS, Chicago, IL, USA). Pearson linear correlation was used to quantify the strength of the relationship between the variables. The results were recognized as statistically significant at a significance level of *p* ≤ 0.05.

## 3. Results

### 3.1. Effectiveness of Submerged (SMF) and Solid-State (SSF) Fermentation of Spirulina

The average pH values of non-fermented samples, i.e., control (I) (Spirulina powder–water mixture (1:20 *w*/*w*)) and control (II) (Spirulina powder–water mixture (1:2 *w*/*w*)), were 6.85 and 6.33, respectively. The pH and viable LAB counts in fermented samples (SMF and SSF) of Spirulina are shown in Figure 2. Among all fermented Spirulina samples, the lowest pH was obtained in samples of 48 h SSF with the No. 173 strain (4.10); however, the highest viable LAB counts were obtained in samples of 24 and 48 h SMF with the No. 225 strain and 48 h SMF and SSF with the No. 245 strain (on average, 9.44 log_10_ CFU/g).

Comparing the results of 24 h fermentation with the same strain and the two types of fermentation (SMF and SSF samples), significant differences in pH values were not established. Significant differences in the viable LAB counts in samples after 24 h of fermentation were also not found. The highest viable LAB counts in Spirulina samples were obtained in SMF and SSF with No. 225 strain (on average, 9.28 log_10_ CFU/g). In the comparison of SMF and SSF samples after 48 h, significant differences in viable LAB counts between groups fermented with No. 244, No. 255 and No. 245 were not found, and viable LAB counts in these samples were, on average, 8.55, 9.17 and 9.47 log_10_ CFU/g, respectively. However, samples fermented with No. 173, after 48 h of SSF, showed higher LAB numbers (on average, 11.4% higher in comparison with SMF samples).

Additionally, the pH values of the Spirulina samples did not differ after 48 h of fermentation with the same LAB strain under different fermentation conditions (SMF or SSF). The effects of the analyzed factors and their interactions on sample pH and LAB count were not significant (Appendix A, Appendix A).

The high viable LAB count and the low pH of the medium are among the most important characteristics of fermented products [36,37]. The fermentation process is strongly influenced by the concentration of fermentable sugars in the substrate [37,38]. The main fermentable carbohydrates in Spirulina are glucose, ribose, galactose, xylose and mannose [39]. Therefore, Spirulina is a suitable material for LAB fermentation, without the need for additional carbon source enrichment [40]. It was reported that the initial pH of *Spirulin*a is, on average, 6.2, and the pH may decline in fermented Spirulina to values as low as 2.9–3.1 because of the organic acids produced during fermentation by LAB, in addition to other metabolites [41,42,43]. In our study, we obtained slightly higher values of pH, as mentioned previously, but it must be emphasized that acidification rates and LAB growth are strain-dependent [44]. The selection of the most appropriate technological starter strain is a critical step in the development of fermented products [44,45]. *Lactiplantibacillus plantarum*, *Lacticaseibacillus casei*, *Lacticaseibacillus rhamnosus* and *Bacillus* strains are popular starters for Spirulina fermentation [40,41,42,43,46] because of their probiotic properties and good technological characteristics for fermenting Spirulina [36,45,47]. The LAB strains used in this study previously showed a good capacity to ferment sugars found in Spirulina [29]. The high numbers of LAB in fermented Spirulina are desirable since they give the product additional probiotic properties [44]. This study showed that the analyzed factors and their interactions did not have statistically significant effects on the viable LAB counts or pH values of Spirulina samples. Finally, according to the results obtained, all of the used LAB strains showed a good capacity to ferment Spirulina without any enrichment with an additional carbohydrate source.

### 3.2. Evaluation of the Concentration of L-Glutamic (L-Glu) and Gamma-Aminobutyric (GABA) Acids in Spirulina Samples

L-Glutamic acid and gamma-aminobutyric acid concentrations in Spirulina samples are presented in Table 1. Comparing all samples, the highest concentration of L-glutamic acid was found in 48 h SSF samples with the No. 173 strain (3841 mg/kg), and this concentration was, on average, 40.2% higher than that found in control (II) samples. Comparing the L-glutamic concentration in 24 h SMF samples with control (I), different tendencies were found: in two sample groups, the L-glutamic acid concentration increased (47.1% on average in SMF samples with No. 244 and No. 225 strains); in contrast, in another two samples, a decrease was observed (93.8% on average in SMF samples with the No. 173 strain, and 90.3% on average in SMF samples with the No. 245 strain). After 48 h of SMF in samples fermented with No. 244 and No. 255, the L-glutamic acid concentration increased by 176 and 22.2% on average, respectively, in comparison with samples after 24 h of fermentation. However, in samples fermented with No. 173 and No. 245 strains, a decrease in the L-glutamic acid concentration was established (10.8 and 1.3 times on average, respectively).

Comparing the L-glutamic acid concentration in 24 h SSF samples with control (II), it was revealed that in three out of four sample groups, the L-glutamic content was higher (in SSF samples with No. 173, No. 225 and No. 245 strains: on average, 43.8, 13.1 and 14.2% higher, respectively), and it was on average 22.3% lower in SSF samples with the No. 244 strain. Additionally, when increasing the duration of fermentation, the same trends were seen. Specifically, in the comparison of L-glutamic acid concentrations after 24 and 48 h SSF with No. 173, No. 225 and No. 245 strains, the L-glutamic acid concentration was on average 16.3, 23.6 and 11.0% higher, respectively, and it was on average 67.9% lower in 48 h SSF samples with the No. 244 strain.

In the comparison of SMF and SSF sample groups, after 24 h of fermentation, higher L-glutamic acid concentrations were found in SSF samples in all cases. The same tendencies were established after 48 h of fermentation.

L-Glutamic acid is a very important brain neurotransmitter largely produced through microbial fermentation [48]. Various microorganisms have the capacity to excrete L-glutamic acid. LAB strains, which are very common microbiological starter cultures in food and feed fermentation [49,50], have a gene responsible for glutamic acid production [51]. Therefore, fermentation with LAB is the most appropriate process for glutamic acid production, because it is considered safe and eco-friendly. However, appropriate LAB strains with desirable metabolic capacities should be selected. Additionally, fermentation conditions constitute a key factor. The current research study showed that the fermentation condition (SMF or SSF) was a significant factor in the L-glutamic acid concentration in Spirulina samples (*p* ≤ 0.0001) (Appendix A, Appendix A). It was reported that glucose is the most appropriate carbon source for glutamic acid production [52], and Spirulina is good source of this sugar [39]. This could be the reason for the observed production of glutamic acid during Spirulina fermentation. It was stated that glutamic acid can be produced by bacteria from glucose via the Krebs cycle [53,54]. Therefore, the presence of glucose in the fermentable substrate is a very important factor for glutamic acid production by LAB. From the above, one may conclude that not only are the fermentation conditions and the type of microbial starters important factors for glutamic acid production, but the use of a substrate rich in glucose is also crucial. Additionally, the pH of the fermentable substrate is important for glutamic acid production [55]. It was reported that the maximum glutamic acid production can be obtained at a lower pH value (4.5) [56]. However, in our study, higher pH values were obtained, and this can be hypothetically explained by the fact that some of the LAB strains can excrete ammonia in an acidic environment, thus contributing to the survival of the microorganisms through pH neutralization [57]. It is important to emphasize that ammonia can reduce glutamic acid production [56]. Yet, it was reported that the production of glutamic acid is mostly dependent on the activity of bacterial cytoplasmic glutamate dehydrogenase [58]. Finally, further investigations are needed to discover the mechanisms involved in glutamic acid production by the LAB strains used in this experiment.

In the comparison of all sample groups, the highest concentration of GABA was found in 48 h SSF samples with the No. 244 strain (on average, 139 times higher than in control (II) samples). Comparing the GABA concentration in the 24 h SMF samples with control (I), in all cases, the GABA concentration in fermented samples increased (in samples with No. 244, No. 173, No. 225 and No. 245, it increased on average by 106, 6, 84.6 and 26.8 times, respectively). Additionally, in most cases, after 48 h of SMF, the GABA concentration increased, except in 48 h SMF samples with the No. 225 strain.

When comparing the GABA concentration in 24 h SSF samples with control (II), we found the same tendencies as those in SMF samples. However, the group of SSF samples with the No. 173 strain showed the lowest GABA content increase—viz., on average 31.4% higher in comparison with control (II).

In the comparison of the GABA concentration after 24 and 48 h of SSF, in sample groups fermented with No. 244 and No. 173 strains, the GABA content increased (on average by 18.8 and 61.6%, respectively), and, in contrast, in sample groups fermented with No. 255 and No. 245 strains, the GABA content decreased (on average by 84.2 and 3.6%, respectively). Moreover, comparing SMF and SSF sample groups after 24 h of fermentation, in all cases, higher GABA concentrations were found in SSF samples. However, after 48 h, different tendencies were observed: in two samples, after 48 h of SSF, the GABA content increased in comparison with SMF samples (in SSF with No. 173 and No. 245 strains), whereas in two samples, the GABA content decreased (in SSF with No. 244 and No. 225 strains).

In contrast to chemical synthesis, biological GABA production using technological microorganisms is safer and more eco-friendly [59,60,61]. There are many LAB species that possess the capacity to produce GABA [28,62,63,64,65,66,67,68,69,70,71,72,73,74,75,76], although the GABA production effectiveness of different LAB strains varies greatly [20]. These tendencies can be seen in our current study as well.

The parameters for the GABA production process can be easily controlled [77]. In technological LAB strains, glucose metabolism produces numerous metabolites, one of which is GABA [78]. However, during this process, GABA can be degraded by γ-aminobutyric acid aminotransferase and semialdehyde dehydrogenase [79]. It was reported that GABA-producing strains were isolated from common fermented food and beverages [80,81,82,83,84,85,86,87]. In this study, LAB were isolated from spontaneous bread sourdough, and some of them showed the potential to produce GABA. Lactic acid bacteria, as economically viable technological microorganisms, are the most studied for GABA production [77]. However, a number of factors (temperature, pH, duration of the process, etc.) can significantly affect the GABA content. It was reported that the optimal temperature for GABA synthesis is 30 °C [87]. However, in another study, the optimal temperature for GABA synthesis was established to be 37 °C [88]. Regarding the optimal pH value, it was found to be 3.5–5 for GABA production by *Lev. brevis* [61], whereas in another study [86], the optimal pH for GABA production by *Enterococcus faecium* was set at 7.74. The influence of pH on GABA production is explained by the optimal pH values for the activity of glutamic acid decarboxylase (GAD) (pH 4.5). In fact, this enzyme in LAB is only active under acidic conditions, and when pH is above 5, GAD loses its activity [8,22]. Note that the optimal pH for fermentation by different LAB strains varies [61,86]. Our study showed that the LAB strain used for fermentation and the fermentation conditions (SMF or SSF), as well as the interaction between these factors, had statistically significant effects on the GABA concentration in Spirulina samples (*p* ≤ 0.001, *p* = 0.019 and *p* = 0.011, respectively). Additionally, taking into consideration that LAB possess not only glutamic acid decarboxylase but also other decarboxylases, BA formation was analyzed in fermented Spirulina samples, because these compounds are usually non-desirable in food, although they can be applied in the pharmaceutical industry.

### 3.3. Evaluation of the Concentrations of Biogenic Amines (BA) in Spirulina Samples

The biogenic amine (BA) concentrations in Spirulina samples are shown in Figure 3, and all of the tested BAs are given in Appendix A in Appendix A. Phenylethylamine was not found in Spirulina samples. Cadaverine was found in 5 samples (control (I), control (II), 24 and 48 h SSF with the No. 244 strain, and 24 h SSF with the No. 225 strain) and histamine was detected in 4 samples (24 and 48 h SSF with the No. 244 strain, 48 h SMF with the No. 173 strain, and 24 h SSF with the No. 225 strain) out of 18 samples (2 controls and 16 fermented samples) (Appendix A, Appendix A).

Cadaverine is formed during the direct decarboxylation of L-lysine through the diaminopimelic acid route in bacteria [89,90]. The direct decarboxylation of L-lysine is catalyzed by lysine decarboxylase in microbial starter cultures [91]. Cadaverine possesses multiple bioactivities [91] and plays a key role in cell survival under acidic conditions [92,93]. Due to its broad functional properties, cadaverine has a huge potential to be applied in agriculture, as well as in medicine [91].

Histamine has been confirmed as cytotoxic [94], and its synergistic effect with tyramine was also recognized [95]. The maximum legal limits of histamine have been established in fish and fish products (200–400 mg/kg, established by the European Union (EU) Commission (EC) Directives 2073/2005 [96], and 50 mg/kg, established by the US Food and Drug Administration (FDA) [97]). During fermentation, histamine is produced by certain LAB, which possess histidine decarboxylase activities [98,99]. The decarboxylation of amino acids is a proton-consuming reaction that may provide acid resistance to some microorganisms [100,101]. These findings suggest that pH is involved in amino acid decarboxylation via enzymatic activity or gene expression [102]. Histamine accumulation is also influenced by other factors, such as temperature, salt concentration, etc. [103,104].

Tryptamine was found in nine out of the sixteen analyzed fermented samples, and its content was below 10 mg/kg of the sample (Figure 3a). However, in all cases, tryptamine was formed in SSF samples (after 24 and 48 h of fermentation). All analyzed factors and their interactions had statistically significant effects on tryptamine formation in Spirulina samples (Appendix A, Appendix A). Tryptamines are medicinally important molecules that serve as precursors to clinically used indole alkaloid natural products [105]. Tryptamine is produced in a single step via tryptophan decarboxylation [99]. The European Food Safety Authority (EFSA) recognizes tryptamine as a potentially harmful BA in foods [106]. At high concentrations, tryptamine can accumulate in fish sauces, certain fish and fish products, dairy products and certain fermented meat products, such as fermented sausages [106]. However, regarding tryptamine accumulation in fermented Spirulina, the data are scarce. Dietary tryptamine can have harmful effects on humans [106,107,108]. Tryptamine can increase the toxicity of histamine [99,109]. The EFSA panel on Biological Hazards (BIOHAZ) highlighted that the lack of knowledge prevents any reliable quantitative or qualitative risk assessment of tryptamine in foods. However, taking into consideration the toxic effect of tyramine, its control in the end product is needed, especially when desirable and non-desirable compound formation is based on the same technological process like it is fermentation with LAB.

In all cases, fermentation increased the putrescine concentration in Spirulina samples, and when comparing all of the samples, the highest putrescine content was found in the 48 h SSF sample with the No. 173 strain (855 mg/kg) (Figure 3b). Contrasting SMF and SSF samples, in all cases, a higher putrescine concentration was found in SSF samples (in the comparison of 24 and 48 h SMF and SSF with the No. 244 strain, 125 and 2.96 times higher on average, respectively; in the comparison of 24 and 48 h SMF and SSF with the No. 173 strain, 9.02 and 8.93 times higher on average, respectively; in the comparison of 24 and 48 h SMF and SSF with the No. 225 strain, 8.11 and 7.85 times higher on average, respectively; and in the comparison of 24 and 48 h SMF and SSF with the No. 245 strain, 3.95 and 6.71 times higher on average, respectively). Moreover, all analyzed factors and their interactions had statistically significant effects on putrescine formation in Spirulina samples (Appendix A, Appendix A). Putrescine is synthesized via ornithine decarboxylation or agmatine deamination [110]. Additionally, it is known that putrescine is able to enhance the toxicological effects of histamine [111]. From another point of view, putrescine is an essential BA to all living organisms and tissues [112,113].

In most cases, fermentation increased the tyramine content in Spirulina samples, except in 24 h SMF samples with No. 173 (in this sample, tyramine was absent) and No. 245 (in this sample, tyramine content was, on average, 7.86 times lower in comparison with control (I) samples) (Figure 3c, Appendix A in Appendix A). Additionally, in most cases, higher tyramine content was formed in SSF samples in comparison with SMF ones (except 48 h SSF samples with No. 173 and No. 245). All analyzed factors and their interactions had statistically significant effects on tyramine formation in Spirulina samples (Appendix A, Appendix A).

Both histamine and tyramine are the most toxic BAs. Furthermore, tyramine is the most abundant BA in fermented foods [114]. The European Food Safety Authority reported that 600 mg/kg tyramine in foods exerts toxic effects on health [106]. Tyramine is generated via the decarboxylation of tyrosine. As in other BAs, this reaction can be influenced by multiple factors, including bacterial activity, the pH of the substrate medium and the salt concentration [115]. The process of fermentation provides particularly high concentrations of tyramine in human nutrition [115]. Tyramine is involved in many physiological processes. However, at high concentrations, it exerts toxic effects, and thus, reliable data about the tyramine content in food and feed are also required. Tyramine-producing bacteria are very popular starters for food and fermentation [116,117,118,119,120]. Currently, commercial starter cultures are evaluated for their capability to generate BAs [106]. On the other hand, technological starters can produce or degrade BAs in fermentable substrate media [121]. The low pH of the substrate enhances tyramine production in a variety of LAB [116,122]. The decarboxylation of amino acids is a cellular mechanism, and the optimal activities of microbial decarboxylases are at acidic pH [123,124,125].

Comparing the spermidine concentrations in Spirulina samples, in all cases, SMF reduced and SSF increased this BA in Spirulina samples (Appendix A, Appendix A). Concerning the spermidine concentrations in 24 and 48 h SMF and SSF samples, in all cases, higher spermidine content was found in SSF samples (in the comparison of 24 and 48 h SMF and SSF with the No. 244 strain, 9.50 and 7.44 times higher on average, respectively; in the comparison of 24 and 48 h SMF and SSF with the No. 173 strain, 9.58 and 9.53 times higher on average, respectively; in the comparison of 24 and 48 h SMF and SSF with the No. 225 strain, 9.26 and 8.95 times higher on average, respectively; and in the comparison of 24 and 48 h SMF and SSF with the No. 245 strain, 8.82 and 7.48 times higher on average, respectively) (Figure 3d). The LAB used for fermentation and the fermentation conditions (SMF or SSF), as well as the interaction between factors (LAB × SMF-SSF and LAB × duration of fermentation × SMF-SSF), had statistically significant effects on the spermidine concentration in Spirulina samples (Appendix A, Appendix A).

Observing the spermine content in Spirulina samples, in most of the SMF samples, spermine was not formed (except in 48 h SMF samples with the No. 245 strain), and this BA content in SSF samples ranged on average between 8.21 mg/kg (in 48 h SSF samples with the No. 225 strain) and 17.5 mg/kg (in the remaining SSF samples) (Figure 3e). Moreover, the LAB used for fermentation and the fermentation conditions (SMF or SSF), as well as the interaction between factors (LAB × SMF-SSF and LAB × duration of fermentation × SMF-SSF), had statistically significant effects on the spermine concentration in Spirulina samples (Appendix A, Appendix A).

Spermidine and spermine have been implicated in the protection against several age-related diseases. Still, increasing their concentrations in the diet is linked to improved health and reduced overall mortality [126]. It is admittedly important for the concentrations of spermidine and spermine in foodstuffs to maintain these BAs at optimal levels in the body [127,128,129]. Spermidine has general antiaging effects [130,131,132,133,134,135,136,137,138,139,140]. Although the contents of polyamines in various types of foods have been reported [127,141,142,143,144,145,146,147,148,149,150,151,152,153,154,155,156,157,158,159,160,161,162,163,164], there is no information about spermidine and spermine concentrations in Spirulina products. Spermidine-rich foods are wheat germ, soybeans, select mushrooms, and various nuts and seeds [148]. Thus, the results of our study may be very important for the data basis for the spermidine and spermine contents in Spirulina products.

In all of the studied cases, higher total BA content was found in SSF Spirulina samples (after 24 and 48 h) than in SMF (in the comparison of 24 and 48 h SMF and SSF with the No. 244 strain, 13.5 and 4.65 times higher on average, respectively; in the comparison of 24 and 48 h SMF and SSF with the No. 173 strain, 9.60 and 7.99 times higher on average, respectively; in the comparison of 24 and 48 h SMF and SSF with the No. 225 strain, 8.95 and 7.10 times higher on average, respectively; and in the comparison of 24 and 48 h SMF and SSF with the No. 245 strain, 6.62 and 6.54 times higher on average, respectively) (Figure 3f).

### 3.4. Antimicrobial Activity of Spirulina Samples

From all of the tested opportunistic and pathogenic strains, fermented Spirulina samples showed exceptional antimicrobial activity against *Staphylococcus aureus* (Table 2). Comparing the diameters of inhibition zones (DIZs) of SMF and SSF samples, in all cases, higher antimicrobial activity was obtained in SSF samples; specifically, the diameters of inhibition zones ranged from 9.2 mm (24 and 48 h SSF samples with the No. 245 strain) to 16.0 mm on average (for the rest of the SSF samples). The LAB strain used for fermentation, as well as the interactions LAB × duration of fermentation, duration of fermentation × SMF-SSF, and LAB × duration of fermentation × SMF-SSF, had statistically significant effects on the diameter of the inhibition zone caused by Spirulina samples against *Staphylococcus aureus* (Appendix A, Appendix A).

Data reported by some authors demonstrated that Spirulina polyphenols, alpha-linolenic acid, C-phycocyanin and the combination of lauric and palmitoleic acids show antimicrobial properties against *Escherichia coli*, *Pseudomonas aeruginosa*, *Bacillus subtilis*, *Aspergillus flavus* and *Aspergillus niger* [165]. Moreover, there are published data regarding the antimicrobial properties of Spirulina methanolic extract against both Gram-positive and Gram-negative pathogens [166]. Additionally, it was reported that the essential oil of *Spirulina platensis* inhibits *S. aureus* (ATCC 25923) and *E. coli* (ATCC 25922), with the most potent effects against *Bacillus antharcis*, *Staphylococcus epidermidis* and *E. coli*, whereas *Salmonella enteritidis* and *P. aeruginosa* (ATCC 27853) were less sensitive to Spirulina essential oils [167]. The antimicrobial activity of the essential oil of Spirulina was explained by the presence of heptadecane, which has a strong antimicrobial effect [168]. The different results of these studies may be explained by the use of different bacterial strains for testing (reference strains vs. wild isolates) or different testing conditions. It seems that the antimicrobial effect of Spirulina compounds is much better expressed against Gram-positive bacteria than Gram-negative bacteria. The results obtained in this study demonstrate a selective antimicrobial effect against *S. aureus*. More interestingly, there was no inhibitory effect on another *Staphylococcus* species—*S. haemolyticus*. Such data suggest that there might be a specific target in *S. aureus* that is affected by Spirulina. However, to confirm such data, further experiments are necessary using more strains of *S. aureus* as well as other *Staphylococcus* species. To date, only a few studies about the antibacterial activity of Spirulina extracts or essential oils have been reported, whereas results about the antimicrobial characteristics of fermented Spirulina are presented in this study for the first time. Other studies are necessary for a better understanding of the antimicrobial properties and mechanisms of fermented Spirulina products.

### 3.5. Relationship between the Formation of Bioactive Compounds of Proteinaceous Origin in Spirulina

The concentrations of bioactive compounds of proteinaceous origin (BA, GABA, L-Glu) in Spirulina samples are given in Figure 4.

Comparing all of the samples, the highest concentrations of GABA were obtained in the SSF samples (24 and 48 h) with the No. 244 strain (higher than 2000 mg/kg) and 24 h SSF samples with the No. 225 strain (1264 mg/kg). Additionally, the BA/GABA ratios in these samples were 0.72, 0.86 and 1.07, respectively. Overall, the BA/GABA ratios in the samples ranged from 0.5 to 62 (in 24 h SMF sample with No. 244 and in 24 h SSF sample with No. 173, respectively). Moreover, the GABA content in Spirulina samples showed significant statistical correlations with putrescine, cadaverine, histamine, tyramine, spermidine and spermine contents (Table 3). The BA/L-Glu ratio in Spirulina samples varied between 0.31 and 10.7 (in 24 h SMF sample with No. 245 and in 48 h SMF sample with No. 244, respectively), and the L-glutamic acid concentration in Spirulina samples showed positive moderate correlations with tryptamine, putrescine, spermidine and spermine (Table 3). Viable lactic acid bacteria counts in Spirulina samples showed weak negative correlations with cadaverine and spermine contents (Table 3). Although viable LAB counts were a significant factor in GABA and L-glutamic acid formation, correlations between them were not established. Furthermore, the GABA concentration in samples showed a weak positive correlation with the diameter of the inhibition zone against *Staphylococcus aureus*. Notwithstanding these results, in this study, correlations between the pH of the samples and the other analyzed parameters were not found.

This study showed that although during the fermentation of Spirulina with LAB, high concentrations of desirable compounds are formed, non-desirable compounds such as BAs are also formed as a result of their similar mechanisms of synthesis, and thus, their eventual presence in high concentrations in the end products must be taken into consideration.

## 4. Conclusions

Spirulina is a suitable substrate for fermentation, and the lowest pH value was obtained in 48 h SSF with the No. 173 Spirulina strain (4.10). The highest viable counts of LAB were acquired in 24 and 48 h SMF samples with the No. 225 strain and 48 h SMF and SSF samples with the No. 245 strain (on average, 9.44 log_10_ CFU/g). The selected LAB strains in this study were shown to possess the capacity to produce L-glutamic acid and GABA in Spirulina biomass (the highest concentration of L-glutamic acid was found in 48 h SSF samples with the No. 173 strain, and the highest concentration of GABA was detected in 48 h SSF samples with the No. 244 strain). In all cases, higher total BA content was found in SSF Spirulina samples when compared with SMF ones. Additionally, fermented Spirulina showed exceptional antimicrobial activity against *Staphylococcus aureus, but not the other tested pathogens*. The biogenic amine/gamma-aminobutyric acid ratio in Spirulina samples ranged from 0.5 to 62, and the BA/L-Glu ratio ranged from 0.31 to 10.7. L-Glutamic acid and GABA contents in Spirulina samples showed significant correlations with some of the identified BAs. Finally, this study showed that, although during fermentation, high concentrations of desirable compounds are formed, non-desirable compounds are also likely to be formed and must be monitored in the end products.

## Figures and Tables

**Figure 1 biology-12-00248-f001:**
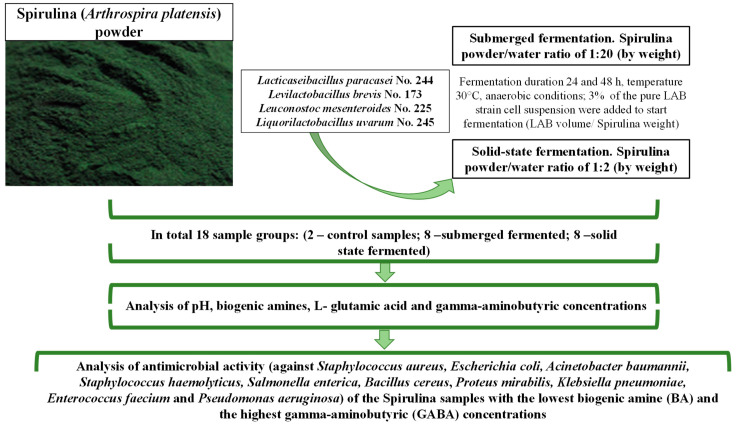
Experimental design.

**Figure 2 biology-12-00248-f002:**
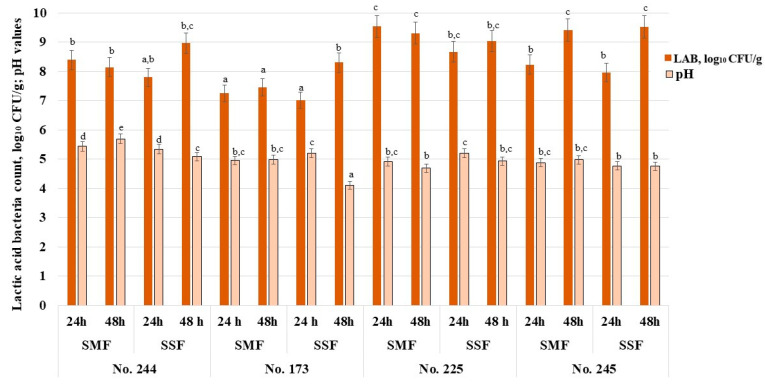
Spirulina pH and viable lactic acid bacteria counts (log_10_ CFU/g). No. 244*—*fermented with *Lacticaseibacillus paracasei* No. 244 strain; No. 173*—*fermented with *Levilactobacillus brevis* No. 173 strain; No. 225—*Leuconostoc mesenteroides* No. 225 strain; No. 245—fermented with *Liquorilactobacillus uvarum* No. 245 strain; LAB—lactic acid bacteria; CFU—colony-forming units; SMF—submerged fermentation; SSF—solid-state fermentation. Data are represented as means (*n* = 6) ± standard errors. ^a–e^ Mean values denoted with different letters indicate significantly different values between the columns (*p* ≤ 0.05).

**Figure 3 biology-12-00248-f003:**
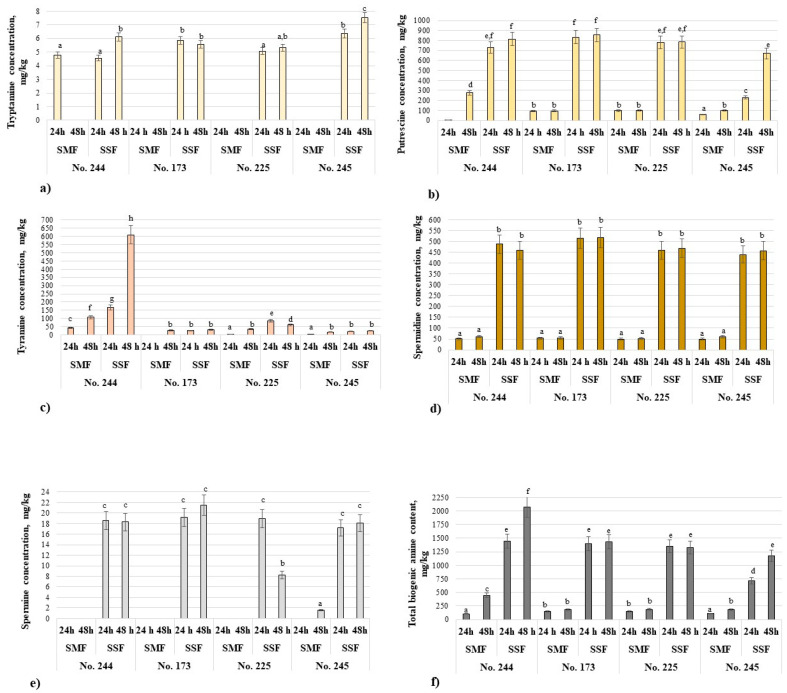
Biogenic amine (BA) content (mg/kg) in Spiulina samples: (**a**) tryptamine, (**b**) putrescine, (**c**) tyramine, (**d**) spermidine, (**e**) spermine, (**f**) total biogenic amine content; No. 244—fermented with *Lacticaseibacillus paracasei* No. 244 strain; No. 173*—*fermented with *Levilactobacillus brevis* No. 173 strain; No. 225—*Leuconostoc mesenteroides* No. 225 strain; No. 245—fermented with *Liquorilactobacillus uvarum* No. 245 strain; LAB—lactic acid bacteria; SMF—submerged fermentation; SSF—solid-state fermentation; ^a–h^ mean values denoted with different letters indicate significantly different values between the columns (*p* ≤ 0.05); data are represented as means (*n* = 6) ± standard errors.

**Figure 4 biology-12-00248-f004:**
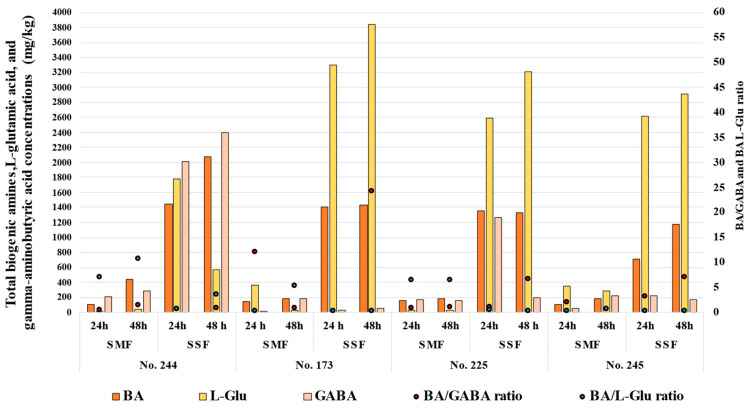
Total biogenic amine (BA), L-glutamic acid (L-Glu) and gamma-aminobutyric acid (GABA) contents (mg/kg) in Spirulina samples and BA/GABA and BA/L-Glu ratios (No. 244*—*fermented with *Lacticaseibacillus paracasei* No. 244 strain; No. 173*—*fermented with *Levilactobacillus brevis* No. 173 strain; No. 225—*Leuconostoc mesenteroides* No. 225 strain; No. 245—fermented with *Liquorilactobacillus uvarum* No. 245 strain).

**Table 1 biology-12-00248-t001:** Concentrations of L-glutamic (L-Glu) and gamma-aminobutyric (GABA) acids in the Spirulina samples.

Spirulina Samples	Fermentation		L-Glutamic Acid, mg/kg	Gamma-Aminobutyric Acid, mg/kg
	Duration, h	Conditions		
Control (I)	-		242 ± 14.8 ^f^	2.01 ± 0.18 ^a^
Control (II)	-		2296 ± 59.3 ^k^	17.2 ± 0.20 ^c^
*Lacticaseibacillus paracasei* No. 244	24 h	SMF	15.1 ± 1.30 ^a^	213 ± 15.0 ^l^
	48 h		41.7 ± 3.20 ^e^	287 ± 21.3 ^h^
	24 h	SSF	1784 ± 24.3 ^j^	2016 ± 46.5 ^j^
	48 h		572 ± 21.2 ^i^	2396 ± 38.6 ^k^
*Levilactobacillus brevis* No. 173	24 h	SMF	357 ± 30.6 ^h^	12.0 ± 0.10 ^b^
	48 h		33.2 ± 2.10 ^d^	187 ± 10.3 ^g^
	24 h	SSF	3302 ± 44.9 ^n^	22.6 ± 1.32 ^d^
	48 h		3841 ± 37.5 ^o^	58.8 ± 4.30 ^e^
*Leuconostoc mesenteroides* No. 225	24 h	SMF	23.4 ± 2.14 ^b^	170 ± 11.2 ^f,g^
	48 h		28.6 ± 2.12 ^c^	162 ± 12.1 ^f^
	24 h	SSF	2597 ± 51.8 ^l^	1264 ± 47.5 ^i^
	48 h		3209 ± 43.0 ^n^	200 ± 15.0 ^g,h^
*Liquorilactobacillus uvarum* No. 245	24 h	SMF	356 ± 15.7 ^h^	53.6 ± 3.1 ^e^
	48 h		280 ± 11.0 ^g^	225 ± 11.2 ^l^
	24 h	SSF	2621 ± 58.4 ^l^	217 ± 8.9 ^l^
	48 h		2908 ± 60.1 ^m^	165 ± 10.1 ^f^

Control (I)—Spirulina powder and water mixture, 1:20 *w*/*w*; control (II)—Spirulina powder and water mixture, 1:2 *w*/*w*; SSF—solid-state fermentation; SMF—submerged fermentation. Data are represented as means (*n* = 6) ± standard errors. ^a–o^ Mean values denoted with different letters indicate significantly different values between the lines (*p* ≤ 0.05).

**Table 2 biology-12-00248-t002:** Diameters (in mm) of the inhibition zones (DIZs) of Spirulina samples.

Spirulina Samples	Fermentation		Pathogenic and Opportunistic Bacterial Strain									
	Dura-tion, h	Condi-tions	*Staphylococcus aureus*	*Escherichia coli*	*Acinetobacter baumannii*	*Staphylococcus haemolyticus*	*Salmonella enterica*	*Bacillus cereus*	*Proteus mirabilis*	*Klebsiella pneumoniae*	*Enterococcus faecium*	*Pseudomonas aeruginosa*
			Diameter of the Inhibition Zone (DIZ), mm									
Control (I)	-		0	0	0	0	0	0	0	0	0	0
Control (II)	-		0	0	0	0	0	0	0	0	0	0
No. 244	24 h	SMF	12.0 ± 0.3 ^c^	0	0	0	0	0	0	0	0	0
	48 h		10.1 ± 0.1 ^b^	0	0	0	0	0	0	0	0	0
	24 h	SSF	16.3 ± 0.2 ^e^	0	0	0	0	0	0	0	0	0
	48 h		15.9 ± 0.3 ^e^	0	0	0	0	0	0	0	0	0
No. 173	24 h	SMF	0	0	0	0	0	0	0	0	0	0
	48 h		13.1 ± 0.1 ^d^	0	0	0	0	0	0	0	0	0
	24 h	SSF	15.9 ± 0.3 ^e^	0	0	0	0	0	0	0	0	0
	48 h		16.1 ± 0.2 ^e^	0	0	0	0	0	0	0	0	0
No. 225	24 h	SMF	0	0	0	0	0	0	0	0	0	0
	48 h		12.2. ± 0.3 ^c^	0	0	0	0	0	0	0	0	0
	24 h	SSF	16.0 ± 0.4 ^e^	0	0	0	0	0	0	0	0	0
	48 h		15.8 ± 0.3 ^e^	0	0	0	0	0	0	0	0	0
No. 245	24 h	SMF	0	0	0	0	0	0	0	0	0	0
	48 h		0	0	0	0	0	0	0	0	0	0
	24 h	SSF	9.1 ± 0.3 ^a^	0	0	0	0	0	0	0	0	0
	48 h		9.3 ± 0.4 ^b^	0	0	0	0	0	0	0	0	0

No. 244*—*fermented with *Lacticaseibacillus paracasei* No. 244 strain; No. 173*—*fermented with *Levilactobacillus brevis* No. 173 strain; No. 225—*Leuconostoc mesenteroides* No. 225 strain; No. 245—fermented with *Liquorilactobacillus uvarum* No. 245 strain; LAB—lactic acid bacteria; SMF—submerged fermentation; SSF—solid-state fermentation; control (I)—Spirulina powder diluted with distilled water (1:20 *w*/*w*) without fermentation; control (II)—Spirulina powder diluted with distilled water (1:2 *w*/*w*) without fermentation; SSF—solid-state fermentation; SMF—submerged fermentation; data are represented as means (*n* = 6) ± standard errors. ^a–e^ Mean values denoted with different letters indicate significantly different values between the samples (*p* ≤ 0.05).

**Table 3 biology-12-00248-t003:** Pearson correlations and their significance between the analyzed Spirulina parameters.

		pH	TRP	PUT	CAD	HIS	TYR	SPRMD	SPRM	GABA	LGlu	DIZ	LAB
pH	r	1	−0.086	−0.197	0.187	0.098	0.053	−0.169	−0.123	0.107	−0.029	0.023	−0.128
	*p*		0.534	0.153	0.175	0.480	0.704	0.222	0.377	0.442	0.833	0.872	0.355
TRP	r	−0.086	1	**0.722 ****	0.255	**0.289 ***	**0.314 ***	**0.877 ****	**0.842 ****	0.256	**0.541****	−0.147	−0.098
	*p*	0.534		**0.0001**	0.062	**0.034**	**0.021**	**0.0001**	**0.0001**	0.061	**0.0001**	0.304	0.482
PUT	r	−0.197	**0.722 ****	1	0.259	**0.377 ****	**0.447 ****	**0.872 ****	**0.747 ****	**0.396 ****	**0.519 ****	0.011	0.073
	*p*	0.153	**0.0001**		0.059	**0.005**	**0.001**	**0.0001**	**0.0001**	**0.003**	**0.0001**	0.938	0.600
CAD	r	0.187	0.255	0.259	1	**0.923 ****	**0.894 ****	0.248	**0.314 ***	**0.531 ****	−0.124	−0.142	**−0.282 ***
	*p*	0.175	0.062	0.059		**0.0001**	**0.0001**	0.070	**0.021**	**0.0001**	0.372	0.322	**0.039**
HIS	r	0.098	**0.289 ***	**0.377 ****	**0.923 ****	1	**0.977 ****	**0.297 ***	**0.306 ***	**0.630 ****	−0.085	−0.029	−0.100
	*p*	0.480	**0.034**	**0.005**	**0.0001**		**0.0001**	**0.029**	**0.024**	**0.0001**	0.539	0.840	0.472
TYR	r	0.053	**0.314 ***	**0.447 ****	**0.894 ****	**0.977 ****	1	**0.325 ***	**0.310 ***	**0.656 ****	−0.065	0.050	−0.0045
	*p*	0.704	**0.021**	**0.001**	**0.0001**	**0.0001**		**0.016**	**0.023**	**0.0001**	0.640	0.727	0.747
SPRMD	r	−0.169	**0.877 ****	**0.872 ****	**0.248**	**0.297 ***	**0.325 ***	1	**0.941 ****	**0.322 ***	**0.627 ****	−0.125	−0.181
	*p*	0.222	**0.0001**	**0.0001**	**0.070**	**0.029**	**0.016**		**0.0001**	**0.018**	**0.0001**	0.383	0.191
SPRM	r	−0.123	**0.842 ****	**0.747 ****	**0.314 ***	**0.30 6***	**0.310 ***	**0.941 ****	1	**0.317 ***	**0.572 ****	−0.133	**−0.347 ***
	*p*	0.377	**0.0001**	**0.0001**	**0.021**	**0.024**	**0.023**	**0.0001**		**0.019**	**0.0001**	0.351	**0.010**
GABA	r	0.107	0.256	**0.396 ****	**0.531 ****	**0.630 ****	**0.656 ****	**0.322 ***	**0.317 ***	1	0.163	**0.337 ***	−0.055
	*p*	0.442	0.061	**0.003**	**0.0001**	**0.0001**	**0.0001**	**0.018**	**0.019**		0.240	**0.016**	0.691
LGlu	r	−0.029	**0.541 ****	**0.519 ****	−0.124	−0.085	−0.065	**0.627 ****	**0.572 ****	0.163	1	−0.099	0.007
	*p*	0.833	**0.0001**	**0.0001**	0.372	0.539	0.640	**0.0001**	**0.0001**	0.240		0.489	0.960
DIZ	r	0.023	−0.147	0.011	−0.142	−0.029	0.050	−0.125	−0.133	**0.337 ***	−0.099	1	0.027
	*p*	0.872	0.304	0.938	0.322	0.840	0.727	0.383	0.351	**0.016**	0.489		0.853
LAB count	r	−0.128	−0.098	0.073	**−0.282 ***	−0.100	−0.045	−0.181	**−0.347 ***	−0.055	0.007	0.027	1
	*p*	0.355	0.482	0.600	**0.039**	0.472	0.747	0.191	**0.010**	0.691	0.960	0.853	

** Correlation is significant at the 0.01 level (2-tailed); * correlation is significant at the 0.05 level (2-tailed); r—Pearson correlation; *p—*significance (2-tailed); LAB—lactic acid bacteria strain used for fermentation; TRP—tryptamine; PHE—phenylethylamine; PUT—putrescine; CAD—cadaverine; HIS—histamine; TYR—tyramine; SPRMD—spermidine; SPRM—spermine; GABA—gamma-aminobutyric acid; LGlu—L-glutamic acid; DIZ—diameter of inhibition zone against *Staphylococcus aureus.*

## Data Availability

Not applicable.

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
