# Peer review of "Submerged and Solid-State Fermentation of Spirulina with Lactic Acid Bacteria Strains: Antimicrobial Properties and the Formation of Bioactive Compounds of Protein Origin"

_biology, 2023, doi:10.3390/biology12020248_

Round 1

Reviewer 1 Report

An interesting topic that requires corrections before being published.

The problem with the title - fermentation was done not only with lactobacilli but also with Leuconostoc. Change the title so that the title of the research follows the objectives and methodology you did. Moreover, the vast majority of references in the Introduction are more than 5 years old, state of the art becomes questionable.

 Correct the abbreviations of bacterial species so that there is no confusion. For example, Enterobacter aerogenes (En. aerogenes), Enterococcus faecium (Ec. faecium), Escherichia coli (E. coli), or Lactobacillus species (L.), Leuconostoc (Ln.), Spirulina platensis, Staphylococcus aureus, and Staphylococcus haemolyticus that they do not start with the same letter.

Reference 194 is very questionable. In that paper (2004), the authors mentioned Streptococcus faecalis ATCC8043 (1. Streptococcus faecalis was renamed into Enterococcus faecalis in 1984, 2. ATCC8043 is Enterococcus hirae). Either remove the reference or modify it to make it unambiguous. Additionally, be careful which scientific papers you cite in the future.

Line 69: Spirulina as Spirulina platensis (if so, italicize) or as the product (provide product characteristics, such as company, country, and city of origin)?

Do not start sentences with abbreviations (Line 118, for example). Correct through the text.

Lines 128-129: change to: with lactobacilli strains (Lacticaseibacillus paracasei No. 244; Levilactobacillus brevis No. 173; Liquorilactobacillus uvarum No. 245) and Leuconostoc mesenteroides No. 225.

Why did you choose these LAB?

Are Spirulina platensis (mentioned in line 68) and Lyophilized Spirulina powder (Arthrospira platensis) the same? if they are, write to be unambiguous.

Is there a fermentation procedure for Spirulina? If there is, please cite the scientific paper.

Line 173 three independent trials?

subheading 2.3. - what was used as a control? Specify. Spirulina powder in water/MRS medium?

subheading 2.4. - what was used as a control? Specify.

subheading 2.5. - Has Spirulina powder in water/MRS medium been used for antimicrobial activity testing? You just wrote pathogenic bacteria, are they ATCC strains, do these strains have any labels? What method did you use to do this?

Figures 2, 3, and 4. I strongly suggest that the figure be better organized, at least as far as colors are concerned. Very uniform deviations with n=6 at CFU, check the raw data. Below Figure 3 there is an explanation of CFU, although there is no such abbreviation anywhere in the figure

Tables 1, 2, and 4: Very confusing, and requires reorganization to be understandable. What do the numbers in the table mean? What does an asterisk mean?

Table 2: Reduce the size of the letters a-o (a-o)

Line 409: Why did you state statistical significance?

Table 5: There is some problem with the table, I cannot see the names of pathogenic bacteria. I assume that the problem arose during the creation of the pdf file and the text breaks. There is an explanation of CFU, although there is no such abbreviation anywhere in the Table. Moreover, why did you not determine the antimicrobial activity against all pathogens (nd)? If there is no activity, do not put nd but 0 mm.

The results and discussion should be better written. The authors tried to discuss the obtained results, however with a reorganization of the sentences this section would be much better.

Author Response

Reviewer 1. An interesting topic that requires corrections before being published.

Authors response: Authors are thankful for valuable comments.

Reviewer 1. The problem with the title - fermentation was done not only with lactobacilli but also with Leuconostoc. Change the title so that the title of the research follows the objectives and methodology you did.

Authors response: Authors are thankful for comment. The title was changed to represent more broader usage of lactic acid bacteria as lactic acid bacteria represent different genera including Lactobacillus and Leuconostoc. The title was changed to:

Submerged and solid-state fermentation of Spirulina with lactic acid bacteria strains: antimicrobial properties and the formation of bioactive compounds of protein origin

Reviewer 1. Moreover, the vast majority of references in the Introduction are more than 5 years old, state of the art becomes questionable.

Authors response: The references in Introduction were updated.

Reviewer 1. Correct the abbreviations of bacterial species so that there is no confusion. For example, Enterobacter aerogenes (En. aerogenes), Enterococcus faecium (Ec. faecium), Escherichia coli (E. coli), or Lactobacillus species (L.), Leuconostoc (Ln.), Spirulina platensisStaphylococcus aureusand Staphylococcus haemolyticus that they do not start with the same letter.

Authors response: corrected.

Reviewer 1. Reference 194 is very questionable. In that paper (2004), the authors mentioned Streptococcus faecalis ATCC8043 (1. Streptococcus faecalis was renamed into Enterococcus faecalis in 1984, 2. ATCC8043 is Enterococcus hirae). Either remove the reference or modify it to make it unambiguous. Additionally, be careful which scientific papers you cite in the future.

Authors response: Authors are thankful for comment, reference was deleted.

Reviewer 1. Line 69: Spirulina as Spirulina platensis (if so, italicize) or as the product (provide product characteristics, such as company, country, and city of origin)?

Authors response: Authors are thankful for comment, information was provided:

Lyophilized Spirulina powder (Arthrospira platensis) (content per 100 g: sodium 1.1 g, total carbohydrates 30.3 g, proteins 60.6 g, calcium 151.5 mg, potassium 1.7 mg, iron 48.5 mg) was provided by Now Foods Company (Illinois, USA).

Reviewer 1. Do not start sentences with abbreviations (Line 118, for example). Correct through the text.

Authors response: corrected.

Reviewer 1. Lines 128-129: change to: with lactobacilli strains (Lacticaseibacillus paracasei No. 244; Levilactobacillus brevis No. 173; Liquorilactobacillus uvarum No. 245) and Leuconostoc mesenteroides No. 225.

Authors response: Authors are thankful for comment, corrected.

Reviewer 1. Why did you choose these LAB?

Authors response: Authors are thankful for comment. These strains, previously isolated from spontaneous fermented cereal, showed different carbohydrate metabolism and desirable antimicrobial properties. However, there are no information, about these LABs application for Spirulina fermentation. We believe, that sourdough LAB application can be much broader, than for bread production. For this reason, these LAB were tested.

Reviewer 1. Are Spirulina platensis (mentioned in line 68and Lyophilized Spirulina powder (Arthrospira platensis) the same? if they are, write to be unambiguous.

Authors response: corrected.

Reviewer 1. Is there a fermentation procedure for Spirulina? If there is, please cite the scientific paper.

Authors response: No, fermentation procedure was performed based on our knowledges in this area.

Reviewer 1. Line 173 three independent trials?

Authors response: Fermentation of the samples was performed in duplicate and all analytical experiments were carried out in triplicate

Reviewer 1.

subheading 2.3. - what was used as a control? Specify. Spirulina powder in water/MRS medium?

subheading 2.4. - what was used as a control? Specify.

Authors response: Non-fermented (non-treated) samples (mixed with sterilized water in appropriate proportions for SMF and SSF) were analyzed as control.

Reviewer 1. subheading 2.5. - Has Spirulina powder in water/MRS medium been used for antimicrobial activity testing? You just wrote pathogenic bacteria, are they ATCC strains, do these strains have any labels? What method did you use to do this?

Authors response:  The sentence was improved: “ Wells with 6 mm in diameter were punched in the agar and filled with 50 µL of the Spirulina samples (mixture of Spirulina powder and sterilized water).”

We have added additional information into the manuscript with explanation that bacterial strains used for testing were wild isolates from human and animal origin, not the reference strains.

Reviewer 1. Figures 2, 3, and 4. I strongly suggest that the figure be better organized, at least as far as colors are concerned. Very uniform deviations with n=6 at CFU, check the raw data. Below Figure 3 there is an explanation of CFU, although there is no such abbreviation anywhere in the figure

Authors response: Authors are thankful for a comment; the colours of figures were improved.

Reviewer 1. Tables 1, 2, and 4: Very confusing, and requires reorganization to be understandable. What do the numbers in the table mean? What does an asterisk mean?

Authors response: Authors are thankful for comment, numbers are p (significance) values (significance value of the factors and their interactions on analysed parameters). Asterisk was changed to „ד. This „ד symbol means interaction between factors.

Reviewer 1. Table 2: Reduce the size of the letters a-o (a-o)

Authors response: corrected.

Reviewer 1. Line 409: Why did you state statistical significance?

Authors response: Authors are thankful for comment, this is important, when different technologies are compared. If Reviewer suggest, we can delete values, because they are given in Supplementary material.

Reviewer 1. Table 5: There is some problem with the table, I cannot see the names of pathogenic bacteria. I assume that the problem arose during the creation of the pdf file and the text breaks. There is an explanation of CFU, although there is no such abbreviation anywhere in the Table. Moreover, why did you not determine the antimicrobial activity against all pathogens (nd)? If there is no activity, do not put nd but 0 mm.

Authors response: Authors are thankful for comment, Table was corrected.

Reviewer 1. The results and discussion should be better written. The authors tried to discuss the obtained results, however with a reorganization of the sentences this section would be much better.

Authors response: The authors are thankful for a valuable comment. We would like to explain that all authors are professionals in their field and confirmed the results and discussion of this manuscript as clearly presented and written; however, some parts of the discussion were improved as requested by the reviewer.

Reviewer 2 Report

The manuscript is very interesting and the topic treated is very topical. However, I feel that the data is shown in a way that is sometimes unusable. In some tables the units of measurement are not present; can some tables be turned into graphs? I would encourage the authors to consider reducing the number of tables, some of which could be included in the supplementary material. Furthermore, in the introduction it would be useful to mention the strengths and weaknesses of the two types of fermentation process SSF and SMF used. Finally, the sampling times carried out and on which the analyzes were then carried out are not described.

Author Response

Reviewer 2. The manuscript is very interesting and the topic treated is very topical. 

Authors response: Authors are thankful for comments.

Reviewer 2. However, I feel that the data is shown in a way that is sometimes unusable. In some tables the units of measurement are not present; can some tables be turned into graphs? I would encourage the authors to consider reducing the number of tables, some of which could be included in the supplementary material. 

Authors response: Authors are thankful for comment. We would like to explain, that these Tables, which are without units, are correct, because in these Tables influence of the analysed factors [lactic acid bacteria (LAB) used for fermentation; duration of fermentation; fermentation conditions] and their interaction on analysed parameters are shown (significant or not significant influence). However, according to Reviewer 2 suggestion, these Tables were included to Supplementary File 2.

Supplementary File 2. Influence of the analysed factors [lactic acid bacteria (LAB) used for fermentation; duration of fermentation; fermentation conditions] and their interaction on analysed parameters: Table S1. Influence of the analysed factors [lactic acid bacteria (LAB) used for fermentation; duration of fermentation; fermentation conditions] and their interaction on lactic acid bacteria viable counts and pH of the Spirulina samples. Table S2. Influence of the analysed factors [lactic acid bacteria (LAB) used for fermentation; duration of fermentation; fermentation conditions] and their interaction on the concentrations of L-glutamic (L-Glu) and gamma-aminobutyric (GABA) acids in Spirulina samples. Table S3. Influence of the analysed factors [lactic acid bacteria (LAB) used for fermentation; duration of fermentation; fermentation conditions] and their interaction on biogenic amine (BA) content in Spirulina. Table S4. Influence of the analysed factors [lactic acid bacteria used for fermentation; duration of fermentation; fermentation conditions) and their interaction on diameter of inhibition zones (DIZ) by Spirulina against Staphylococcus aureus.

Table S1. Influence of the analysed factors [lactic acid bacteria (LAB) used for fermentation; duration of fermentation; fermentation conditions] and their interaction on lactic acid bacteria viable counts and pH of the Spirulina samples.

Spirulina parameters

LAB

Duration of fermentation

SMF-SSF

LAB × Duration of fermentation

LAB × SMF-SSF

Duration of fermentation × SMF-SSF

LAB × Duration of fermentation × SMF-SSF

pH

0.724

0.524

0.066

0.527

0.246

0.124

0.795

LAB viable counts

0.154

0.481

0.067

0.439

0.533

0.284

0.693

LAB – lactic acid bacteria strain used for fermentation; SMF – submerged fermentation; SSF – solid-state fermentation; Factor or factors interaction is significant, when p ≤ 0.05

Table S2. Influence of the analysed factors [lactic acid bacteria (LAB) used for fermentation; duration of fermentation; fermentation conditions] and their interaction on the concentrations of L-glutamic (L-Glu) and gamma-aminobutyric (GABA) acids in Spirulina samples.

Spirulina parameters

LAB

Duration of fermentation

SMF-SSF

LAB × Duration of fermentation

LAB × SMF-SSF

Duration of fermentation × SMF-SSF

LAB × Duration of fermentation × SMF-SSF

GABA

0.001

0.868

0.019

0.561

0.011

0.497

0.503

L-Glu

0.368

0.910

0.0001

0.889

0.433

0.937

0.771

GABA – gamma-aminobutyric acid; LGlu – L-glutamic acid; LAB – lactic acid bacteria strain used for fermentation; SMF – submerged fermentation; SSF – solid-state fermentation; Factor or factors interaction is significant, when p ≤ 0.05.

Table S3. Influence of the analysed factors [lactic acid bacteria (LAB) used for fermentation; duration of fermentation; fermentation conditions] and their interaction on biogenic amine (BA) content in Spirulina.

 Spirulina parameters

LAB

Duration of fermentation

SMF-SSF

LAB × Duration of fermentation

LAB × SMF-SSF

Duration of fermentation × SMF-SSF

LAB × Duration of fermentation × SMF-SSF

TRP

0.001

0.021

0.0001

0.001

0.0001

0.0001

0.0001

PUT

0.0001

0.0001

0.0001

0.0001

0.0001

0.0001

0.0001

CAD

0.0001

0.0001

0.0001

0.0001

0.0001

0.0001

0.0001

HIS

0.0001

0.0001

0.0001

0.0001

0.0001

0.0001

0.0001

TYR

0.0001

0.0001

0.0001

0.0001

0.0001

0.0001

0.0001

SPRMD

0.0001

0.370

0.0001

0.082

0.0001

0.229

0.020

SPRM

0.0001

0.263

0.0001

0.0001

0.0001

0.001

0.001

LAB – lactic acid bacteria strain used for fermentation; SMF – submerged fermentation; SSF – solid-state fermentation; TRP – tryptamine; PHE – phenylethylamine; PUT – putrescine; CAD – cadaverine; HIS – histamine; TYR – tyramine; SPRMD – spermidine; SPRM – spermine; Factor or factors interaction is significant, when p ≤ 0.05.

Table S4. Influence of the analysed factors [lactic acid bacteria used for fermentation; duration of fermentation; fermentation conditions) and their interaction on diameter of inhibition zones (DIZ) by Spirulina against Staphylococcus aureus.

 Spirulina parameters

LAB

Duration of fermentation

SMF-SSF

LAB × Duration of fermentation

LAB × SMF-SSF

Duration of fermentation × SMF-SSF

LAB × Duration of fermentation × SMF-SSF

DIZ

0.0001

0.209

0.514

0.0001

0.822

0.0001

0.0001

DIZ – diameter of inhibition zones; LAB – lactic acid bacteria strain used for fermentation; SMF – submerged fermentation; SSF – solid-state fermentation; Factor or factors interaction is significant, when p ≤ 0.05.

Reviewer 2. Furthermore, in the introduction it would be useful to mention the strengths and weaknesses of the two types of fermentation process SSF and SMF used. 

Authors response: Authors are thankful for comment, an explanation was included:

Solid-state fermentation (SSF) consists of the microbial growth and product formation on solid particles in the absence (or near absence) of water. This technology is more economical, compared with the traditional way of biomass cultivation in a liquid medium containing nutrients.

Reviewer 2. Finally, the sampling times carried out and on which the analyzes were then carried out are not described.

Authors response: Authors are thankful for comment, please see in section 2.1.:

Afterwards, the algae were fermented under anaerobic conditions in a chamber incubator (Memmert GmbH Co. KG, Schwabach, Germany) for 24 and 48 h, at 30 °C. Non-fermented (non-treated) samples (mixed with sterilized water in appropriate proportions for SMF and SSF) were analyzed as control.
